# Formation of multinucleated osteoclasts depends on an oxidized species of cell surface-associated La protein

Evgenia Leikina[1†], Jarred M Whitlock[1*†‡], Kamran Melikov[1], Wendy Zhang[1], Michael P Bachmann[2,3,4], Leonid Chernomordik[1*]

[1]Section on Membrane Biology, Eunice Kennedy Shriver National Institute of Child Health and Human Development, National Institutes of Health, Bethesda, United States; [2]University Cancer Center (UCC), Tumor Immunology, University Hospital Carl Gustav Carus Dresden, Technical University Dresden, Dresden, Germany; [3]Department of Radioimmunology, Institute of Radiopharmaceutical Cancer Research, Helmholtz-Zentrum Dresden-Rossendorf (HZDR), Dresden, Germany; [4]Institute of Immunology, Medical Faculty Carl Gustav Carus Dresden, Technical University Dresden, Dresden, Germany

**For correspondence:**
aes4xx@virginia.edu (JMW);
chernoml@mail.nih.gov (LC)

†These authors contributed equally to this work

**Present address:** ‡University of Virginia, Charlottesville, United States

**Competing interest:** The authors declare that no competing interests exist.

**Abstract** The bone-resorbing activity of osteoclasts plays a critical role in the life-long remodeling of our bones that is perturbed in many bone loss diseases. Multinucleated osteoclasts are formed by the fusion of precursor cells, and larger cells – generated by an increased number of cell fusion events – have higher resorptive activity. We find that osteoclast fusion and bone resorption are promoted by reactive oxygen species (ROS) signaling and by an unconventional low molecular weight species of La protein, located at the osteoclast surface. Here, we develop the hypothesis that La's unique regulatory role in osteoclast multinucleation and function is controlled by an ROS switch in La trafficking. Using antibodies that recognize reduced or oxidized species of La, we find that differentiating osteoclasts enrich an oxidized species of La at the cell surface, which is distinct from the reduced La species conventionally localized within cell nuclei. ROS signaling triggers the shift from reduced to oxidized La species, its dephosphorylation and delivery to the surface of osteoclasts, where La promotes multinucleation and resorptive activity. Moreover, intracellular ROS signaling in differentiating osteoclasts oxidizes critical cysteine residues in the C-terminal half of La, producing this unconventional La species that promotes osteoclast fusion. Our findings suggest that redox signaling induces changes in the location and function of La and may represent a promising target for novel skeletal therapies.

## eLife assessment

This manuscript provides an **important** advance in our understanding of the molecular events that promote osteoclast fusion. **Compelling** data support the conclusion that an oxidized form of the ubiquitous protein La promotes osteoclast fusion following enrichment at the cell surface of osteoclast progenitors. These data improve our understanding of the processes that regulate bone resorption and will be of broad interest to researchers in the fields of cell biology and musculoskeletal physiology.

## Introduction

The integrity of our bones throughout life depends on tightly regulated coordination between the bone-forming activity of osteoblasts and the bone-resorbing activity of osteoclasts (*Kim et al., 2020*; *McDonald et al., 2021*; *Bolamperti et al., 2022*). A variety of genetic and age-related skeletal

disorders are linked to a disbalance in osteoblast–osteoclast functional coupling that commonly results in excessive bone resorption and/or insufficient synthesis and mineralization of bone.

Multinucleated osteoclasts are formed by the fusion of mononucleated precursor cells, and, in most cases, cells with more nuclei (i.e., generated by a larger number of fusion events) have higher resorptive activity (*Piper et al., 1992*; *Møller et al., 2020*). We recently demonstrated that both osteoclast fusion and bone resorption are controlled by an unconventional, low molecular weight form of La protein (*Whitlock et al., 2023*). La (SSB small RNA-binding exonuclease protection factor La, Gene ID 6741, NCBI GENE), an abundant, ubiquitous protein in eukaryotes, exists primarily as a phosphorylated, nuclear species that plays essential functions in the maturation of RNA polymerase III transcripts, particularly tRNA (*Wolin and Cedervall, 2002*; *Maraia et al., 2017*). However, we discovered that La is dephosphorylated, proteolytically cleaved, and delivered to the surface of osteoclasts during multinucleation. Surface-associated La promotes cell–cell fusion and increased resorptive capacity in osteoclasts. When osteoclasts reach a mature size appropriate for their biological activity, fusion stops. Completion of the fusion process coincides with the removal of surface-associated La, suggesting that the changes in the surface La amounts effectively set the size and biological activity of osteoclasts. The mechanisms that trigger the radical switch in the location and function of La from nucleus to cell surface; and from ubiquitous RNA chaperone to an osteoclast-specific fusion regulator – remain to be understood.

Intracellular reactive oxygen species (ROS) represents a common biological switch for proteins with multiple functions in eukaryotes (*Dishman and Volkman, 2018*; *Liu and Jeffery, 2020*). Excessive levels of ROS during oxidative stress eventually lead to irreversible damage of biological systems and tissues and have been linked to diverse diseases in many physiological systems, including the skeleton (*Forman and Zhang, 2021*). However, transient, moderate increases in ROS levels, referred to as redox signaling (*Schieber and Chandel, 2014*; *Lennicke and Cochemé, 2021*) or a mild oxidative stress (*Ďuračková, 2010*), play important roles in diverse cellular differentiation processes (*Forman et al., 2010*; *Wilson, 2014*; *Domazetovic et al., 2017*). Intracellular ROS signaling commonly induces the formation of disulfide bonds, drives structural and oligomeric transitions, and promotes the unconventional secretion of some proteins lacking a signal peptide (*Urano et al., 2018*; *Kwak et al., 2019*; *Cruz-Garcia et al., 2020*). Alternatively, a transition in the redox state of cysteine residues can also be triggered merely by protein trafficking changes that shifts the localization of a protein from the cytosol (typically reducing) to the extracellular environment (typically oxidizing) (*Gilbert, 1990*; *Ottaviano et al., 2008*).

Like many other physiological processes, bone remodeling and, more specifically, osteoclast formation depend on ROS signaling (*Domazetovic et al., 2017*; *Wang et al., 2011*). Receptor activator of NF-kappaB ligand (RANKL)-induced differentiation of osteoclast precursors quickly generates transient ROS signaling in differentiating osteoclasts (*Lee et al., 2005*), and many bone diseases, including osteoporosis, have been linked to perturbations in ROS signaling (*Domazetovic et al., 2017*; *Reis and Ramos, 2021*). Moreover, application of oxidizing reagents, such as $H_2O_2$, promotes osteoclast formation (*Suda et al., 1993*; *Agidigbi and Kim, 2019*; *Bartell et al., 2014*; *Lean et al., 2005*; *Garrett et al., 1990*). In contrast, cell-permeable antioxidants block RANKL-induced ROS production and inhibit osteoclast formation and bone resorption (*Huh et al., 2006*; *Sanders et al., 2007*; *Cao and Picklo, 2014*; *Kim et al., 2019*).

Recent biochemical studies demonstrate that oxidizing conditions and intracellular redox signaling elicit conformational transitions and oligomerization of La protein and promote its nucleus-to-cytoplasm shuttling (*Berndt et al., 2021a*; *Berndt et al., 2021b*). Here, we tested the hypothesis that La's unique regulatory role in osteoclast multinucleation and function is controlled by an ROS switch in La trafficking and function. Using antibodies that recognize reduced vs oxidized species of La, we found that nuclear La and cell surface La in differentiating osteoclasts to represent reduced and oxidized species of the protein, respectively. Oxidized La species at the surface of osteoclasts promoted their fusion and increased multinucleation. Suppressing ROS signaling in osteoclast precursors inhibited the appearance of La in the cytoplasm and at the surface of osteoclasts and suppressed fusion during osteoclast formation. Addition of the C-terminal half of La – the region required for promoting osteoclast fusion (*Whitlock et al., 2023*) – to the extracellular surface of osteoclasts rescued this inhibition but only if critical La cysteine residues were available for oxidation. Our data suggest that transient ROS signaling induces a shift from reduced to oxidized La species and plays a critical role in directing

the delivery of La to the surface of osteoclasts and promoting multinucleation and subsequent resorptive function of these syncytial bone remodelers. Our findings suggest that redox transition and, more specifically, La cysteine oxidation may represent promising targets for therapeutic strategies aimed at modulating osteoclast-dependent bone resorption in skeletal pathologies.

## Results

### Osteoclast fusion depends on an oxidized form of surface La

To mechanistically evaluate the transition from osteoclast precursors to multinucleated osteoclasts, we incubated primary human monocytes with recombinant macrophage colony-stimulating factor (M-CSF) and then with M-CSF and recombinant RANKL (*Whitlock et al., 2023*; *Asagiri and Takayanagi, 2007*). While the time course and efficiency of multinucleated osteoclast formation vary from donor to donor (*Whitlock et al., 2023*), on average, we observed the appearance of small, multinucleated syncytia 2 days after RANKL application and fusion rapidly increased over the next 2 days, resulting in mature, resorption competent osteoclasts at 4–5 days post-RANKL addition (*Whitlock et al., 2023*). Previously, we demonstrated that the cell fusion stage of multinucleated osteoclast formation depends on La trafficking to the surface of the osteoclasts at days 2–4 post-RANKL application (*Whitlock et al., 2023*); however, the mechanisms that trigger La's trip to the osteoclast surface and what molecular requirements must be met for its unconventional fusion role there remained open questions.

To address open questions concerning La's surface trafficking and molecular function in osteoclast multinucleation, we evaluated the redox state of La in fusing osteoclasts using recently validated monoclonal α-La antibodies that recognize oxidized La (clone 7B6) or reduced La (clone 312B), or do not distinguish between these La species (Pan, clone 5B9) (*Berndt et al., 2021a*). At the time of fusion, immunofluorescence analysis of La's localization in permeabilized osteoclasts with pan α-La antibody demonstrated that La is present in both the cytoplasm and nuclei (*Figure 1a*). In stark contrast, the cytoplasmic/plasma membrane-associated pool was recognized by the α-La antibody that recognizes oxidized La species, while the α-La antibody that recognizes reduced La species recognized the nuclear protein pool (*Figure 1a*; *Figure 1—figure supplement 1*). Under non-permeabilizing conditions – focusing on the exofacial surface of the plasma membrane – we readily observed La at the surface of fusing osteoclasts (see also *Whitlock et al., 2023*). We find that this surface pool of La is dramatically enriched in oxidized, rather than reduced, La species (*Figure 1b*). In further support of this conclusion, we found that application of the membrane-impermeable reducing reagent Tris (2-carboxyethyl) phosphine (TCEP) dramatically decreased the recognition of surface La by the α-La antibody that recognizes the oxidized species and increased the recognition of surface La by the α-La antibody that recognizes the reduced species (*Figure 1b, c*). In contrast, TCEP had no effect on the ability of the pan α-La antibody to recognize surface La (*Figure 1b, c*) or on the surface detection of La with another α-La antibody (Abcam #75927) that we have used previously to evaluate La at the surface of fusing osteoclasts (*Whitlock et al., 2023*) and now found to recognize both reduced and oxidized species of La (data not shown). These findings indicate that the surface pool of La that manages osteoclast size and resorptive function is primarily composed of an oxidized La molecular species. Interestingly, TCEP treatment did not dissociate cell surface La from plasma membrane, suggesting that the reduced species of La retains its ability to associate with the plasma membrane.

We further assessed the functional importance of surface La's redox status in synchronized osteoclast fusion. As in earlier studies (*Whitlock et al., 2023*; *Verma et al., 2014*; *Verma et al., 2018*), we uncoupled the cell fusion stage of osteoclast formation from pre-fusion differentiation processes using lysophosphatidylcholine (LPC), a reversible inhibitor of an early stage of membrane rearrangement required for osteoclast fusion. LPC was applied for 16 hr following 2 days of RANKL elicited osteoclastogenesis. Ready-to-fuse cells, which could not fuse in the presence of LPC, rapidly fused after LPC wash out (*Figure 2a, b*). As we demonstrated previously with another pan α-La antibody (*Whitlock et al., 2023*), pan α-La antibody 5B9 applied at the time of LPC removal inhibited synchronized osteoclast fusion (*Figure 2a, b*). Similarly, α-La antibodies that recognize the oxidized species inhibited fusion, while antibodies that recognize the reduced La species had no effect on fusion (*Figure 2a, b*). These data strongly support the conclusion that the functional La species that promotes osteoclast fusion and function is an unconventional, cell surface-associated, oxidized species. In further support of the functional importance of surface La oxidation, and possibly other surface proteins, we find that

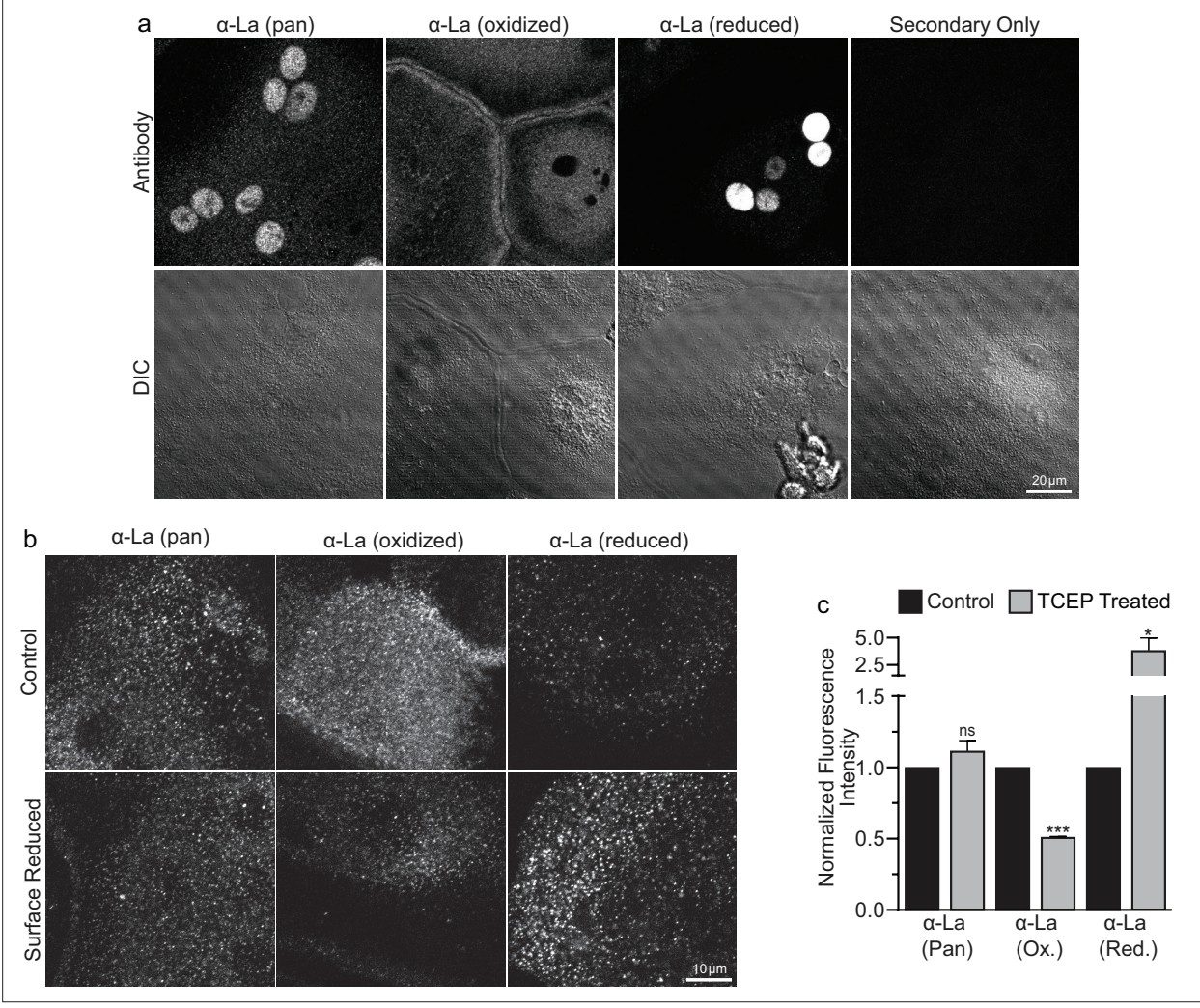

**Figure 1.** Oxidized La decorates the surface of osteoclasts during multinucleation. (**a**) Representative immunofluorescence (top) and differential interference contrast (DIC, bottom) confocal micrographs of permeabilized primary human osteoclasts. La localization was visualized via a general α-La antibody (Pan), an α-La antibody that recognizes oxidized La, or an α-La antibody thar recognizes reduced La (described and validated in *Berndt et al., 2021b*). (**b**) Representative immunofluorescence confocal micrographs of primary human osteoclasts stained with the antibodies described in (**a**) under non-permeabilized conditions to visualize surface La. Surface La pools were visualized for the untreated cells (control) and for cells treated with the membrane-impermeable reducing reagent TCAP. (**c**) Quantification of (**b**) (n = 3) (p = 0.13, 0.0001, and 0.03, respectively). Statistical significance evaluated via paired *t*-test. * = p < 0.05. *** ≤ p < 0.0001. Data are presented as mean values +/- SEM.

The online version of this article includes the following source data and figure supplement(s) for figure 1:

**Source data 1.** Tabular data for *Figure 1c*.

**Figure supplement 1.** Oxidized La is found in the cytosol.

reducing the surface of human osteoclasts via TCEP treatment – which reduces surface La but does not alter its membrane association (*Figure 1c*) – inhibits synchronized osteoclast fusion in a dose-dependent manner (*Figure 2c*).

To further validate the role of La's redox status in osteoclast multinucleation and resorptive function, we chose to assess the functional impact of perturbing the oxidation status of recombinant La. Previously, we showed that the C-terminal half of La produced recombinantly and added to the medium bathing differentiating osteoclast precursors binds the surface of these cells and is sufficient to promote osteoclast fusion and resorptive function (*Whitlock et al., 2023*). We produced La 194–408 recombinantly and reduced a portion of the protein with TCEP followed by the application of iodoacetamide to block the free thiols in La 194–408, preventing their subsequent oxidation. When we compared the effects of La 194–408 vs reduced La 194–408 on primary human osteoclast

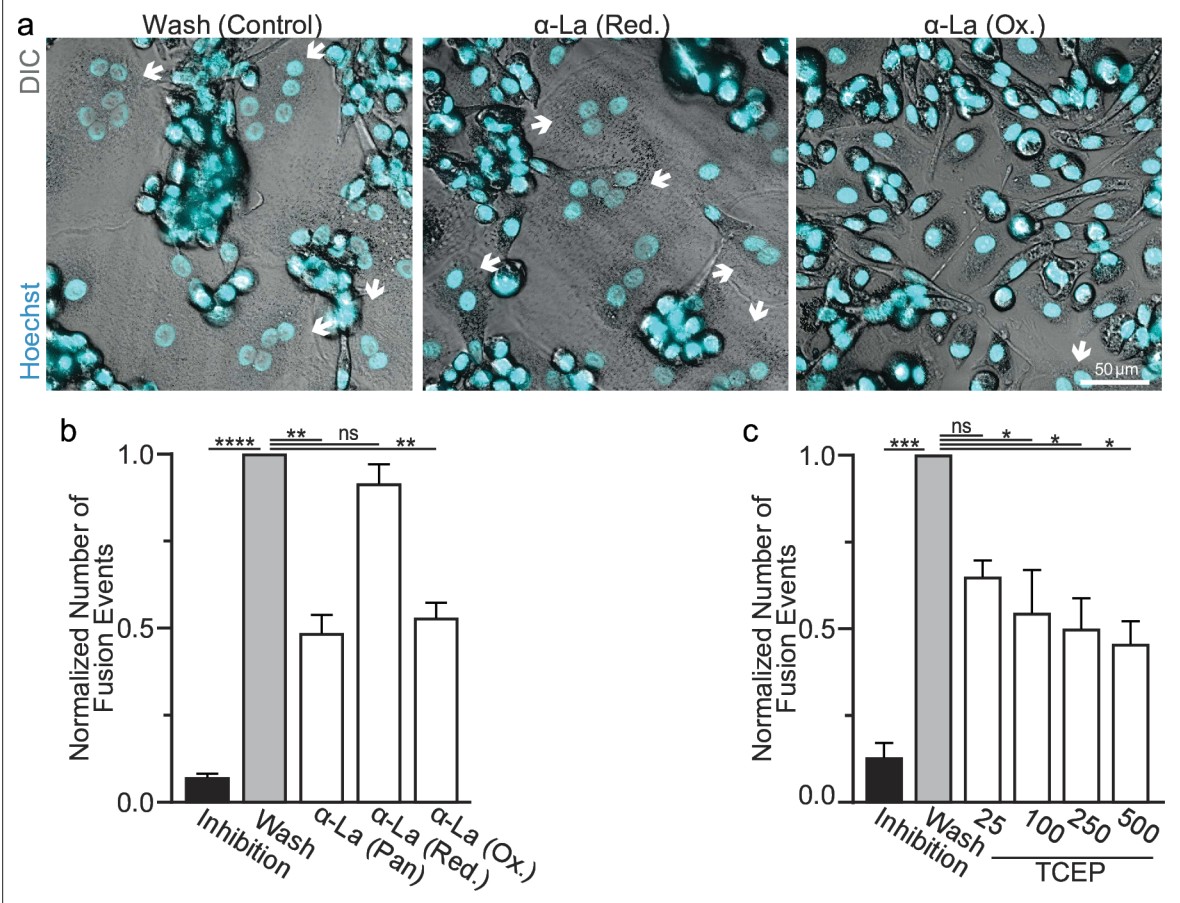

**Figure 2.** Oxidized La promotes osteoclast membrane fusion. (**a**) Representative fluorescence and differential interference contrast (DIC) confocal micrographs of primary human osteoclasts following synchronized cell–cell fusion where hemifusion inhibitor was left (Inhibition), removed (Wash), or removed but the α-La antibodies indicated were simultaneously added. Cyan = Hoechst arrows = multinucleated osteoclasts. (**b**) Quantification of (**a**) (*n* = 5) (p = <0.0001, 0.0019, 0.44, and 0.0038, respectively). (**c**) Quantification of synchronized primary human osteoclast fusion events under control conditions or conditions where surface proteins are reduced (Tris (2-carboxyethyl) phosphine, TCEP). Osteoclast fusion was synchronized by reversibly inhibiting cell–cell fusion using the membrane fusion inhibitor lysophosphatidylcholine (LPC). Inhibition = LPC applied and not removed, Wash = LPC applied and removed to allow synchronized fusion, TCEP = same as Wash with the addition of TCEP before LPC removal (n = 4, except 25 where n = 2) (p = 0.0005, 0.36, 0.02, 0.016, and 0.013, respectively). Statistical significance evaluated via paired one-way analysis of variance (ANOVA) with Holm–Sidak correction. * = p < 0.05, ** = <0.01, *** = p < 0.001, ***** ≤ p < 0.0001. Data are presented as mean values +/- SEM.

The online version of this article includes the following source data for figure 2:

**Source data 1.** Tabular data for *Figure 2b*.

**Source data 2.** Tabular data for *Figure 2c*.

multinucleation, we found that the reduction of recombinant La greatly diminished its ability to promote osteoclast multinucleation and resorption (*Figure 3a–c*).

Many of the effects of redox signaling are mediated by the oxidation of cysteine residues within proteins (*van der Reest et al., 2018*). Human La has three cysteine residues. However, we have previously demonstrated that the N-terminal half of La (La 1–187), containing one of La's cysteines, is dispensable for La's role in promoting osteoclast fusion (*Whitlock et al., 2023*). Therefore, we focused on the two cysteines in the C-terminal half of La: Cys 232 and Cys 245. We recombinantly produced a La 194–408 mutant where cysteines 232 and 245 were mutated to alanine residues (La Cys Mut) (*Figure 4a, b*; *Figure 4—figure supplement 1a*). Interestingly, we find that a fraction of La 194–408 migrates as a band ~twice the predicted size of La 194–408 despite separation via denaturing gel electrophoresis, suggesting the presence of La 194–408 dimers (*Berndt et al., 2021a*; *Craig et al., 1997*). In contrast, we observe no dimer band in La Cys Mut (*Figure 4—figure supplement 1b, c*). In addition, we find that the loss of Cys 232 and Cys 245 greatly diminishes the affinity of the α-La

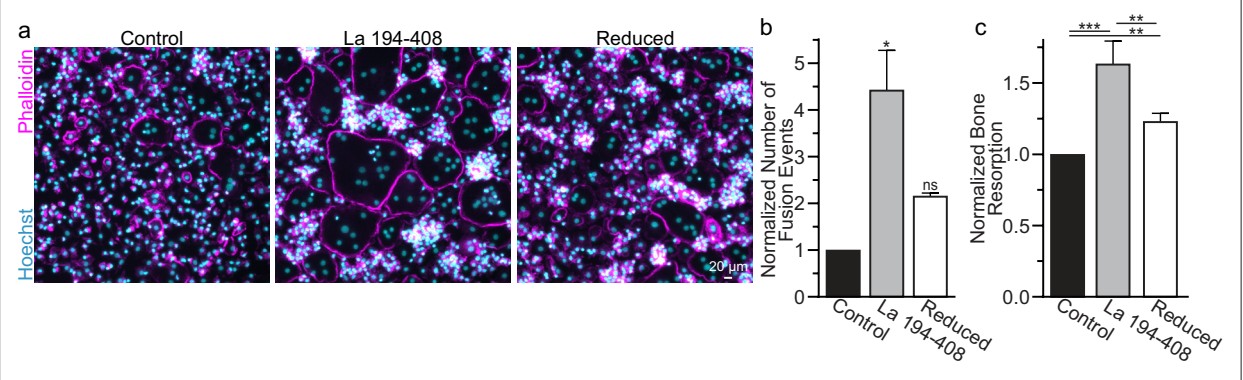

**Figure 3.** Surface La's oxidation is functionally important for the formation and function of osteoclasts. (**a**) Representative fluorescence images of primary human osteoclast multinucleation following addition of control La 194–408 vs La 194–408 where cysteine residues were reduced by Tris (2-carboxyethyl) phosphine (TCEP) and then blocked by iodoacetamide treatments. Cyan = Hoechst. Magenta = phalloidin. (**b**) Quantification of osteoclast fusion events in (**a**) ($n = 3$) ($p = 0.029$ and $0.44$, respectively). Statistical significance evaluated via paired Friedman test with Dunn's correction. (**c**) Quantification of in vitro resorptive function in conditions described in (**a**) ($n = 4$) ($p = 0.0001$, $0.009$, and $0.001$, respectively). Statistical significance evaluated via paired one-way analysis of variance (ANOVA) with Holm–Sidak correction. * = $p < 0.05$, ** = $< 0.01$, *** = $p < 0.001$. Data are presented as mean values +/- SEM.

The online version of this article includes the following source data for figure 3:

**Source data 1.** Tabular data for *Figure 3b*.

**Source data 2.** Tabular data for *Figure 3c*.

antibody that recognizes the oxidized La species, whereas both species are recognized by an α-6xHis monoclonal antibody via their N-terminal His tags (*Figure 4b*). Finally, we found that the loss of Cys 232 and Cys 245 strongly abrogates the ability of La 194–408 to promote osteoclast fusion (*Figure 4c, d*). From these data, we conclude that Cys 232 and Cys 245 – previously highlighted for their roles in redox-dependent structural changes in La (*Berndt et al., 2021a*) – are vital for La's function in osteoclast multinucleation.

To summarize, La promotes osteoclast fusion as an unconventional, cell surface-associated, oxidized species. Moreover, our data suggest that the conformational transition from reduced to oxidized La species that is critical for the protein's role in osteoclast formation and function depends on cysteine residues within its C-terminal half.

## La's redox transition takes place in the cytoplasm and promotes surface delivery

The La pool within the nuclei of eukaryotic cells is primarily a reduced species (*Berndt et al., 2021a*; *Berndt et al., 2021b*). Finding that the surface La pool, which promotes multinucleation in osteoclasts, is enriched in an oxidized species raised the question of where La becomes oxidized? Is La oxidation in osteoclasts a consequence of surface delivery or does La oxidation precede surface trafficking?

Finding that La in permeabilized differentiating osteoclasts is recognized by the α-La antibody that recognizes oxidized rather than reduced La species (*Figure 1a* and *Figure 1—figure supplement 1*) suggested that the oxidation of La takes place in the cytoplasm of forming osteoclasts prior to delivery to the oxidizing environment of their surface. To further address this question, we utilized a cell-permeable reducing reagent *N*-acetylcysteine (NAC) that has been widely used to inhibit intracellular ROS generation and suppress cytosolic redox signaling (*Zafarullah et al., 2003*; *Sun, 2010*). Since under some conditions complex indirect effects of NAC raise rather than lower ROS levels (*Pedre et al., 2021*; *Mlejnek et al., 2021*; *Murphy et al., 2022*), we verified the ROS-lowering effects of NAC in osteoclast precursors using a previously validated, cell-permeable ROS probe CellROX Deep Red (*Lee et al., 2005*). We found that NAC treatment suppresses ROS elevation stimulated by RANKL-initiated differentiation of human osteoclasts (*Figure 5a, b*). While early treatment with NAC has been reported to disrupt the pre-fusion stages of osteoclast differentiation (*Lee et al., 2005*), we found that NAC treatment 24 hr after RANKL application (i.e., after early osteoclast precursor commitment), did not lower the steady-state levels of osteoclastogenic differentiation factors (NFATc1 and

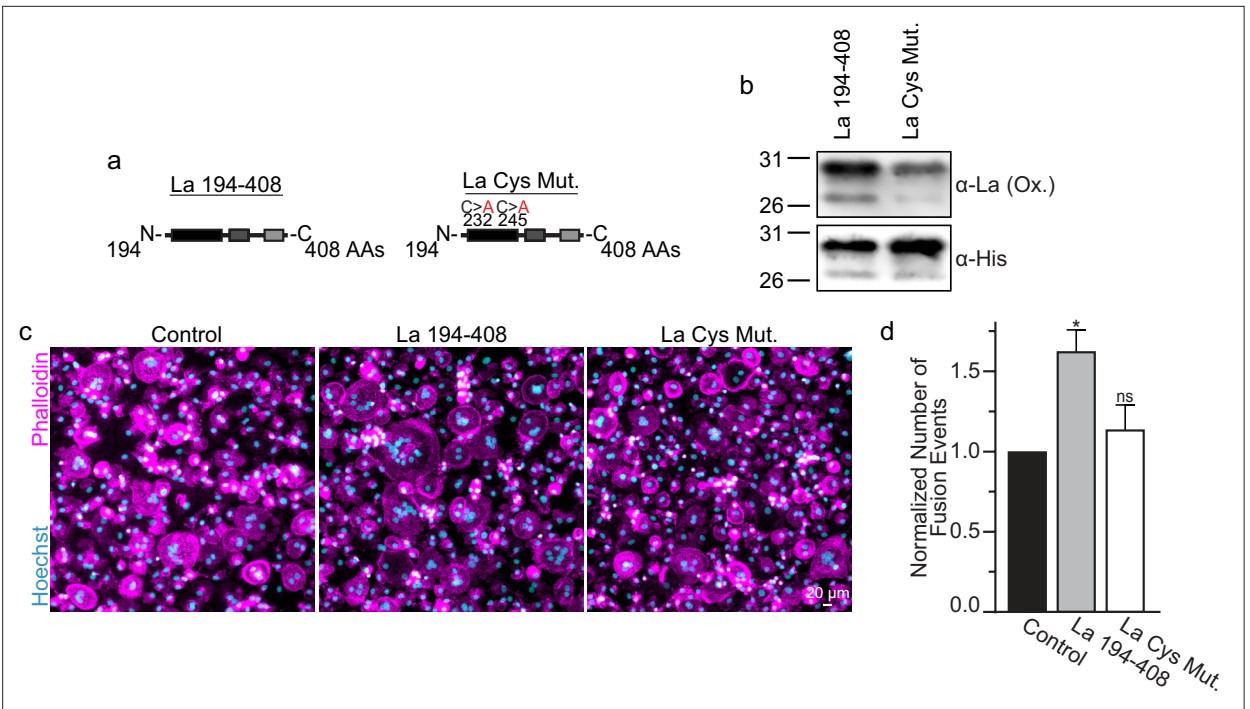

**Figure 4.** La 194–408 cysteine residues are critical for promoting osteoclast fusion. (**a**) Cartoons illustrating the domain structure of La's C-terminal half and the location of its two cysteines. (**b**) Representative Western Blots depicting La C-terminal half and cystine mutant. (**c**) Representative immunofluorescence micrographs of fusing primary human osteoclasts under control conditions or treated with La 194–408 or cysteine mutant La 194–408. (**d**) Quantification of the number of fusion events observed in (**c**) (n = 4) (p = 0.046 and 0.62, respectively). Statistical significance evaluated via paired one-way analysis of variance (ANOVA) with Dunnett correction. * = p < 0.05. Data are presented as mean values +/- SEM.

The online version of this article includes the following source data and figure supplement(s) for figure 4:

**Source data 1.** Gel images for *Figure 4b*.

**Source data 2.** Raw gel images for *Figure 4b*.

**Source data 3.** Raw tabular data for *Figure 4d*.

**Figure supplement 1.** La C-terminal half and cysteine mutant purification.

**Figure supplement 1—source data 1.** Gel images.

**Figure supplement 1—source data 2.** Raw gel images.

cFOS) or the transcripts of two fusion-related proteins (La or AnxA5) (*Figure 5c*). We then explored the effects of suppressing ROS signaling on osteoclast formation and the trafficking of La, as in *Cao and Picklo, 2014*; *Kim et al., 2019*, we found that NAC-mediated inhibition of ROS signaling suppresses osteoclast multinucleation in a dose-dependent manner (*Figure 5d*). Suppressing ROS signaling with NAC also inhibited the transition from reduced to oxidized, cytoplasmic La species, as evidenced by decrease in staining of permeabilized osteoclasts with α-oxidized La antibody and a complementary increase in staining of permeabilized osteoclasts with α-reduced La antibody (*Figure 5—figure supplement 1a–d*). Most importantly, NAC application inhibited the cell surface delivery of La in human osteoclasts (*Figure 5e*). These findings suggest that NAC inhibits La function in osteoclasts by suppressing ROS-induced oxidation of La and its delivery to the surface of osteoclasts. Moreover, our findings also indicate that the transition from reduced to oxidized La species occurs inside the osteoclast cytoplasm during their commitment program rather than being a consequence of La arriving at the oxidizing environment of the extracellular milieu.

While most of La in human cells is phosphorylated at Ser 366 (*Schwartz et al., 2004*), at the time of fusion, La is mostly dephosphorylated, as evidenced by a loss of the cell staining with antibodies specific for La phosphorylated at Ser366 (α-p366 La Ab) (*Whitlock et al., 2023*). We find that while little phosphorylated La is typically observed in the nuclei of differentiating osteoclasts during fusion timepoints, that NAC suppression of transient ROS signaling greatly increases phosphorylated La in

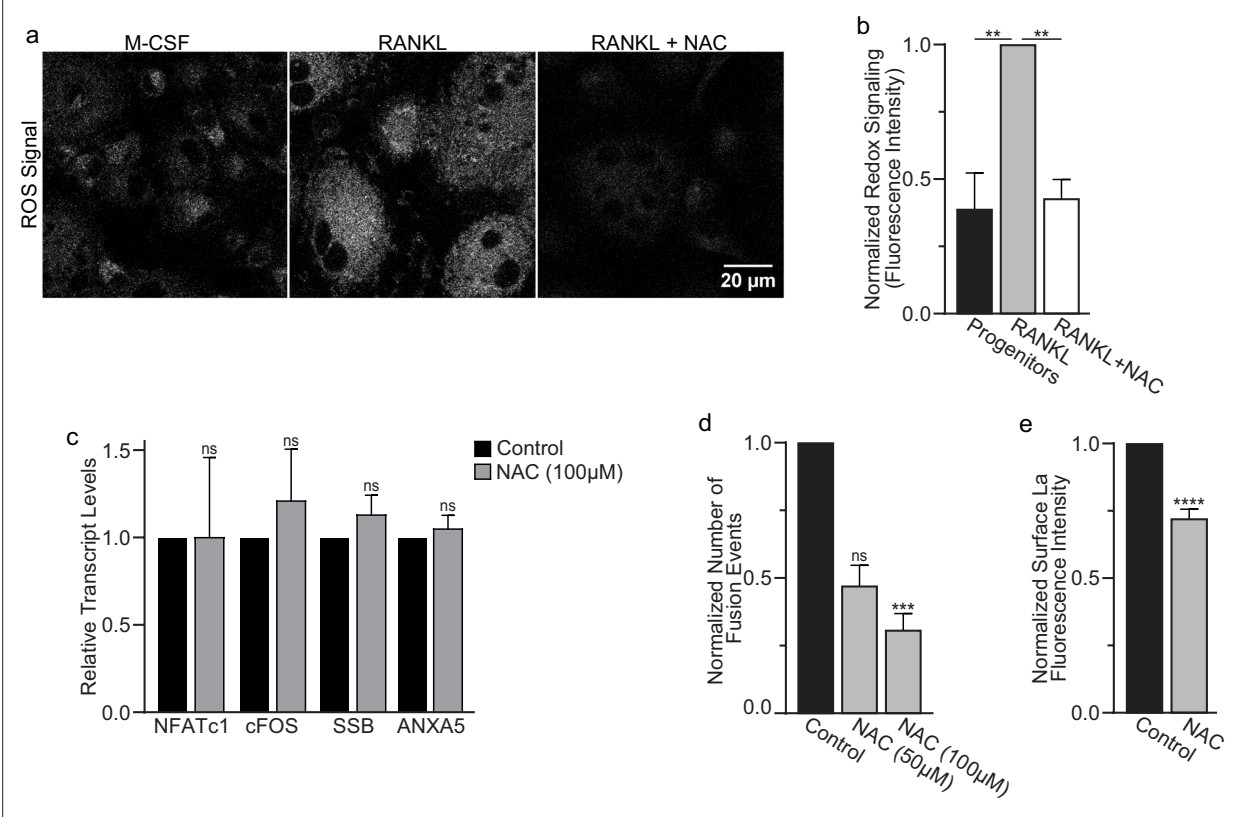

**Figure 5.** Reactive oxygen species (ROS) promotes oxidized La surface trafficking and osteoclast fusion. (**a**) Representative confocal micrographs of ROS signal in primary human osteoclasts precursors under conditions lacking receptor activator of NF-kappaB ligand (RANKL), following 16 hr of RANKL application, or following 16 and 1 hr 100 μM *N*-acetylcysteine (NAC) treatment (gray = CellRox Dye). (**b**) Quantification of ROS signaling in osteoclast progenitors, committed osteoclasts, or committed osteoclasts treated with the membrane-permeable reducing reagent NAC (n = 2, 4, and 3, respectively) (p = 0.004 and 0.0043, respectively). Statistical significance evaluated via paired one-way analysis of variance (ANOVA) with Holm–Sidak correction. (**c**) qPCR quantification of osteoclastogenesis markers (*NFATc1* and *cFOS*), La transcript (*SSB*), and annexin A5 (*ANXA5*). Expression evaluated in comparison to *GAPDH* (n = 4) (p = 0.50, 0.69, 0.32, and 0.45, respectively). Statistical significance evaluated via paired *t*-test. (**d**) Quantification of the number of fusion events observed between human osteoclasts in control conditions or conditions where fusion was inhibited via NAC treatment (n = 6) (p = 0.057 and 0.019, respectfully). Statistical significance evaluated via paired one-way ANOVA with Holm–Sidak correction. (**e**) Quantification of La surface staining of non-permeabilized cells with pan α-La antibodies at day 3 post-RANKL application without or with 50–100 μM NAC added at day 1 post-RANKL application (n = 9) (p = <0.0001). Statistical significance evaluated via paired *t*-test. ** = <0.01, *** = p < 0.001 ***** ≤ p < 0.0001. Data are presented as mean values +/- SEM.

The online version of this article includes the following source data and figure supplement(s) for figure 5:

**Source data 1.** Raw tabular data for *Figure 5b*.

**Source data 2.** Raw tabular data for *Figure 5c*.

**Source data 3.** Raw tabular data for *Figure 5d*.

**Source data 4.** Raw tabular data for *Figure 5e*.

**Figure supplement 1.** *N*-Acetylcysteine (NAC) inhibits redox shift from reduced to oxidized species of intracellular La.

**Figure supplement 1—source data 1.** Raw tabular data for *Figure 5—figure supplement 1b*.

**Figure supplement 1—source data 2.** Raw tabular data for *Figure 5—figure supplement 1c*.

osteoclasts (*Figure 6a, b*). These data indicate that inhibition of intracellular ROS generation prevents La dephosphorylation and the loss of nuclear localization in differentiating osteoclast precursors. These findings substantiate the hypothesis that the redox signaling, which triggers a shift in La functional properties, is important for osteoclast fusion and takes place inside the cell.

We found that NAC application both inhibits osteoclast multinucleation and suppresses delivery of oxidized La species to the cell surface of fusing osteoclasts (*Figure 5d, e*). These findings motivated us to explore whether the suppressed osteoclast multinucleation caused by NAC treatment could be

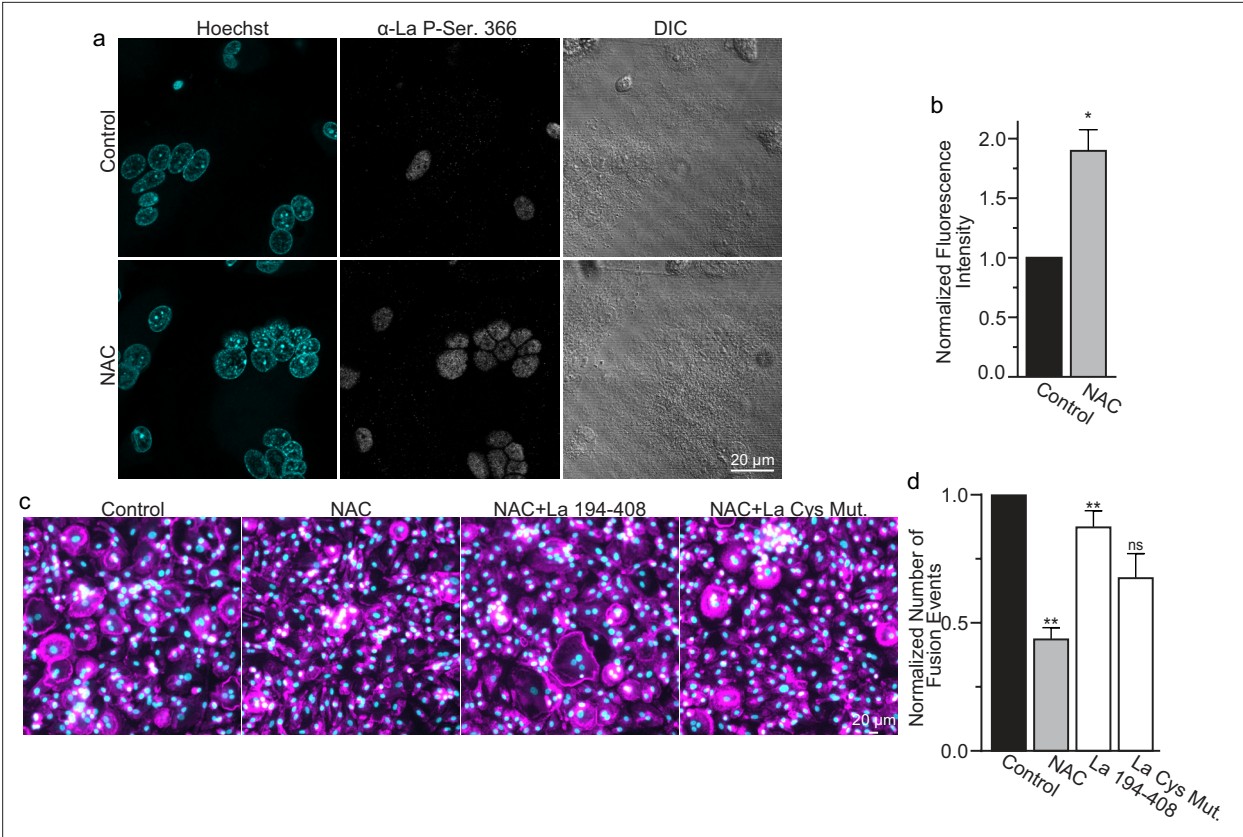

**Figure 6.** Reactive oxygen species (ROS) signaling promotes dephosphorylation of La and osteoclast fusion by increasing the amounts of oxidized La at the surface of the cells. (**a**) Fluorescence microscopy and differential interference contrast (DIC) images of permeabilized osteoclasts without or with application of 100 mm *N*-acetylcysteine (NAC) (1 hr) stained with an α-La antibody that recognizes La phosphorylated at Ser366. (**b**) Quantification of the staining intensity from (**a**) (n = 3) (p = 0.01). Statistical significance evaluated via paired *t*-test. (**c**) Representative fluorescence images of differentiating osteoclasts in control conditions, conditions where fusion was inhibited via NAC treatment (100 µM NAC added at 2 days post-receptor activator of NF-kappaB ligand [RANKL] application), and conditions where fusion was rescued by the application of recombinant La 194–408 or cysteine mutant La 194–408. (**d**) Quantification of the number of osteoclast fusion events in (**c**) (n = 5) (p = 0.0007, 0.0001, 0.0054, and 0.0352, respectfully). Statistical significance for Control vs NAC or La 190–408 vs La Cys Mutant was evaluated via paired *t*-test (p = 0.0012 and 0.0037, and 0.073, respectively). Statistical significance for NAC vs La 190–408 or La Cys Mutant rescue was evaluated via one-way analysis of variance (ANOVA) with Holm–Sidak correction. * = p < 0.05, ** = <0.01. Data are presented as mean values +/- SEM.

The online version of this article includes the following source data for figure 6:

**Source data 1.** Raw tabular data for *Figure 6b*.

**Source data 2.** Raw tabular data for *Figure 6d*.

the result of deficient La delivery to the cell surface. If the suppressed fusion observed following NAC treatment is a consequence of suppressing an ROS triggered 'La trafficking switch', then perhaps simply adding La to the medium bathing NAC-treated cells can rescue La surface pools and fusion? Indeed, we found that application of La 194–408 to NAC-treated cells rescued osteoclast fusion inhibition (*Figure 6c, d*). However, the ability of La 194–408 to rescue NAC-inhibited osteoclast fusion is at l=east partially dependent on Cys 232 and Cys 245, as mutation of these residues to Ala greatly diminished the ability of La 194–408 to rescue NAC effects on fusion. Finding that the NAC-mediated ROS suppression of surface trafficking and fusion can be compensated for by the application of exogenous La strongly supports the conclusion that ROS signaling plays a vital role in triggering La's delivery to the surface of osteoclasts and the promotion of their multinucleation. Moreover, the loss of La's ability to rescue NAC suppression of osteoclast multinucleation when Cys 232 and Cys 245 are mutated to Ala also suggests that these residues are vital for oxidized La's ability to promote osteoclast multinucleation and resorptive function.

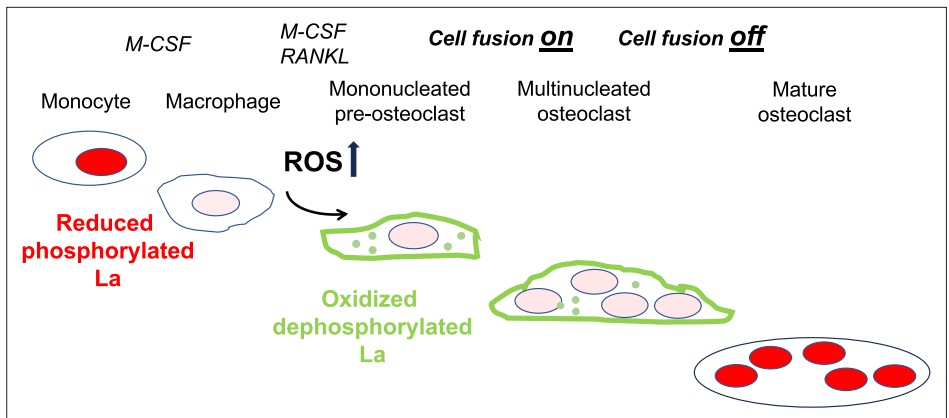

**Figure 7.** Reactive oxygen species (ROS) signaling induced restructuring of La from reduced to oxidized species triggers La re-localization from nucleus to the surface of differentiating osteoclasts and promotes their fusion and resorptive function. An illustrated depiction of osteoclastogenic differentiation from monocytes to mature bone-resorbing osteoclasts. Machrophage precursors are derived via the macrophage colony-stimulating factor (M-CSF) activation of circulating monocytes. Osteoclastst differentiation is initiated by subsequent application of M-CSF and receptor activator of NF-kappaB ligand (RANKL), which ellicits intracellular ROS production leading to drastic changes in the redox state and localization of La. La transitions from a prodominatly nuclear, reduced species of La in monocytes and macrophages to an oxidized, dephosphorylated species that traffics to and associates with the surface of fusion-competent osteoclasts. When osteoclast arrive at an appropriate size and fusion stops, the mature multinucleated osteoclasts exhibit a predominately nuclear, reduced La species, as is typical of other eukaryotic cells.

In summary, ROS signaling downstream of osteoclast commitment contributes to the cell fusion stage of osteoclast formation by promoting intracellular oxidation of La and its delivery to the surface of fusion-committed cells, where La promotes multinucleation and subsequent resorptive activity in human osteoclasts (*Figure 7*).

## Discussion

The reversible shift of redox homeostasis in cells to a moderately oxidized state, referred to as redox signaling, regulates the function and localization of many proteins and favors differentiation vs proliferation (*Smith et al., 2000*; *Hansen and Harris, 2015*; *Tan et al., 2017*). Our findings confirm earlier reports emphasizing the importance of redox signaling in osteoclastogenic differentiation (*Domazetovic et al., 2017*; *Wang et al., 2011*; *Lee et al., 2005*). ROS signaling and its inhibition by NAC likely influence osteoclast formation in many ways and at many points in the formation of multinucleated osteoclasts (*Lee et al., 2005*). However, our finding that exogenous La rescues the NAC-mediated suppression of osteoclast fusion supports the hypothesis that ROS signaling promotes cell fusion stage of osteoclastogenesis by triggering La's delivery to the surface of osteoclasts. This mechanism enriches our recent identification of La protein as a regulator of the cell–cell fusion stage of osteoclast formation (*Whitlock et al., 2023*) and highlights the role of osteoclast ROS signaling in directing fusion machinery to the surface of osteoclast progenitors. While the reduced species of La functions as an essential nuclear RNA chaperone, oxidized La shifts to the cytoplasm and shuttles to the surface of the osteoclasts. It is this specialized oxidize species of cell surface-associated La that promotes osteoclast fusion and bone resorption.

The transition from reduced to oxidized species of La is accompanied by many changes in the protein structure, including an oxidation-induced oligomerization of the protein, a loss of almost half of its helical content, and a change in accessibility of epitopes recognized by conformation specific antibodies (*Berndt et al., 2021a*). Our data show that La belongs to a category of fold switching proteins that change functions in response to the cellular environment, more specifically, oxidizing and reducing intracellular conditions (*Kim and Porter, 2021*). In addition to direct changes in the structure of La caused by its oxidation, ROS can influence La function in osteoclast fusion by indirect effects. Formation of multinucleated osteoclasts involves the caspase 3-cleaved, non-phosphorylated species

of La (*Whitlock et al., 2023*) and ROS can influence La properties by activation of both caspase 3 (*Liu et al., 2019*) and PP2A-like phosphatase (*Cicchillitti et al., 2003*), previously reported to dephosphorylate La Ser366 before La cleavage (*Rutjes et al., 1999*). Indeed, while our finding that NAC inhibits dephosphorylation of La can indicate that redox-dependent changes in La's conformation are required for the dephosphorylation of the protein, it can also be explained by the ROS dependence of PP2A activity. In both scenarios, redox signaling promotes intracellular La modifications and trafficking that delivers the fusion-promoting species of La to the surface of cells.

We still do not know how cell surface La promotes osteoclast fusion and why the oxidized species of La is functional in this unique cellular context. For many viral and intracellular fusion processes, remodeling of membrane bilayers in fusion is thought to be driven by the conformational energy released at the time and place of fusion in the restructuring of fusion-promoting proteins (*Weber et al., 1998*). Finding that La acquires an oxidized conformation already in cytoplasm rather than at the cell surface at the time of fusion argues against the hypothesis that redox restructuring of La directly contributes to membrane remodeling. Furthermore, our finding that La association with the cell surface does not change after reducing oxidized La with TCEP argues against the hypothesis that La oxidation is merely required for its association with the surface of osteoclasts.

Our data, which indicate that fusion competence in osteoclast precursors depends on ROS signaling, can be, at least partially, explained by the oxidation-dependent changes in the structure of La protein and its nucleo-cytoplasmic-cell surface shuttling. Formation of disulfide bonds; re-localization of normally nuclear proteins to the cytoplasm, extracellular medium, and cell surface; and dramatic changes in function in response to transient increases in ROS concentrations have been well described for other nuclear chaperones, in particular, high mobility group Box 1 (HMGB1) (*Kwak et al., 2019*; *Chen et al., 2022*; *Yang et al., 2022*). Like La, HMGB1 has three cysteines and mild HMGB1 oxidation generates disulfide bonds that stabilize homodimers (*Kwak et al., 2020*). Homodimerization by formation of disulfide bonds is also a pre-requisite for the unconventional secretion of Fibroblast Growth Factor 1 (*Prudovsky et al., 2008*).

The specific pathway(s), by which RANKL-induced increases in ROS levels (*Lee et al., 2005*) shift the redox state of La and facilitate its unconventional secretion, remain to be clarified. Moreover, we hope to explore the contributions of related pathways like the production of nitric oxide (NO) and other reactive nitrogen species. In the case of HMGB1, the secretion of the protein, allowing it to act as a signaling molecule outside the cell, in addition to ROS, depends on NO signaling mediated by NO binding of one of a cysteine thiol in protein (*Yang et al., 2022*). Interestingly, like redox signaling (*Domazetovic et al., 2017*; *Wang et al., 2011*; *Lee et al., 2005*), NO signaling promotes osteoclastogenesis (*Nilforoushan et al., 2009*) and the translocation of La protein to the cytoplasm (*Berndt et al., 2021b*). More work is needed to explore the contributions of NO signaling to La function in osteoclasts and to compare the molecular mechanisms that deliver these two redox-, fold-, and function-shifting proteins, La and HMGB1, to the cell surface and extracellular medium.

In conclusion, in this study, we identified redox signaling as a molecular switch that redirects La protein away from the nucleus, where it protects precursor tRNAs from exonuclease digestion, and toward its separable function at the osteoclast surface, where La regulates the multinucleation and resorptive functions of these managers of the skeleton.

Proteins involved in osteoclastogenesis represent potential therapeutic targets for treating bone loss diseases. Finding that osteoclast La promotes bone resorption by acting not only at a different location but also in a different conformation from in comparison to the species of La that carries out its essential and ubiquitous RNA chaperoning functions (cell surface vs nucleus and oxidized vs reduced form), may help in minimizing off target effects of La targeting treatments.

## Methods
### Reagents
Human M-CSF and RANKL were purchased from Cell Sciences (Catalogue # CRM146B and CRR100B, respectively). LPC (1-lauroyl-2- hydroxy-sn-glycero-3-phosphocholine, # 855475); PC (1,2-dioleoyl-sn-glycero-3-phosphocholine, # 850375 C) was purchased from Avanti Polar Lipids. Bone Resorption Assay Kits were purchased from Cosmo Bio Co (Catalogue # CSR-BRA-24KIT) and used according to the manufacturer's instructions. Hoechst 33342 and phalloidin-Alexa 555 were purchased from

Invitrogen (# H3570 and A30106, respectively). TCEP was purchased from Thermo Fisher Scientific (Pierce TCEP-HCl; Catalogue # A35349). NAC (*N*-acetyl-L-cysteine) and sodium iodoacetamide were purchased from Sigma-Aldrich (Catalogue # A9165 and GERPN6302).

## Cells

Elutriated monocytes from healthy donors were obtained through the Department of Transfusion Medicine at National Institutes of Health under protocol 99-CC-0168 approved by the National Institutes of Health Institutional Review Board. Research blood donors provided written informed consent and blood samples were de-identified prior to distribution, Clinical Trials Number: NCT00001846. We also used elutriated monocytes from healthy donors obtained through Elutriation Core Facility, University of Nebraska Medical Center, informed consent was obtained under an Institutional Review Board approved protocol for human subject research 0417-22-FB. Research blood donors provided informed consent and samples were de-identified prior to distribution. Primary human osteoclasts were derived as described previously (*Whitlock et al., 2023*). Briefly, elutriated monocytes were added to complete media [α-minimal essential media (α-MEM) (Gibco) + 10% fetal bovine serum (FBS) (Gibco) + 1× penicillin/streptomycin/glutamate (Gibco)] supplemented with 100 ng/ml recombinant M-CSF and plated at $1 \times 10^6$ cells/ml for 6 days (refreshing media at day 3). Next, the cells were placed into complete medium supplemented with 100 ng/ml recombinant M-CSF and 100 ng/ml recombinant RANKL to induce osteoclastogenesis 3–4 days to obtain multinucleated, resorption competent human osteoclasts.

## Antibodies

Murine monoclonal α-La antibodies that specifically recognize oxidized and reduced species of La or both species of La (7B6, 312B, and 5B9, respectively *Berndt et al., 2021a*), referred to as oxidized La Ab, reduced La Ab, and pan α-La Ab were described and characterized by the laboratory of Dr. Michael Bachmann. The antibodies were produced recombinantly as described in *Berndt et al., 2021a*.

We also used rabbit anti-La Phospho-Ser366 antibody (Abcam, 61800), referred to as α-p366 La Ab that recognizes phosphorylated human La (phosphoSer366). We also used an additional monoclonal murine α-La antibody (α-La mAb; Abcam, Catalogue # 75927) that we found to recognize both oxidized and reduced forms of La. A α-6xhis murine monoclonal antibody (Abcam, ab18184) was used to recognize the 6xhis tag covalently modifying our recombinantly produced La protein fragments and α-Cyclophilin B (Cell Signaling Technology, D1VdJ Rabbit mAb #43603) as a loading control.

## Constructs and recombinant protein

Constructs encoding recombinant La 194–408 and the cysteine mutants used in this manuscript were previously described, characterized, and provided by the Bachmann Lab. Each was transformed into BL2 (DE3) chemically competent *Escherichia coli* (Thermo Fisher Scientific) and recombinant protein production was induced via isopropyl-beta-D-thiogalactoside induction (Sigma). Cells were lysed with BugBuster HT (Millipore) supplemented with protease inhibitors (Complete, Pierce), 6xHis-La proteins were purified using HisPur Cobalt Spin columns (Thermo Fisher Scientific), and endotoxin was removed via Pierce high-capacity endotoxin removal columns (Thermo Fisher Scientific), each according to the manufacturer's instructions. Proteins were sterile filtered, aliquoted, and kept at −80°C. Some La 194–408 was irreversibly modified via 45-min incubation with 10 mM TCEP followed by a 45-min incubation with 10 mM iodoacetamide. Modified protein was then subsequently exchanged into phosphate buffered saline (PBS) (Gibco) using 10 K MW concentrators according to the manufacturer's instructions (Amicon). Control La 194–408 for these experiments was treated identically, except for the omission of TCEP and iodoacetamide.

## Microscopy

For high-resolution immunofluorescence analysis of protein localization, we washed cells with PBS and then rapidly fixed with warm, freshly prepared 4% formaldehyde in PBS at 37°C. The cells were subsequently washed with PBS. To permeabilize cells, we incubated them for 5 min in 0.1% Triton X-100 in PBS. The cells were subsequently stained in PBS supplemented with 10% FBS for 10 min at room temperature to suppress non-specific binding. Then, cells were incubated with primary antibodies for

1 hr in PBS supplemented with 10% FBS. After five washes in PBS, we incubated the cells with fluorescent secondary FAB fragments raised to the species corresponding to the primary antibody for 1 hr in PBS supplemented with 10% FBS (either Anti-rabbit IgG Fab2 Alexa Fluor 555 or Anti-mouse IgG Fab2 Alexa Flour 488, both Cell Signaling Technology, Catalogue # 647 4414 S and 4408 S, respectively, in 1:500 dilution) and then washed five times with PBS prior to imaging. For non-permeabilized conditions, we followed the same protocol, but omitted all use of detergents.

Images were captured on a Zeiss LSM 800, confocal microscope using a C-Apochromat 63×/1.2 water immersion objective lens.

For basic immunofluorescence analysis of cell fusion and morphology, we stained osteoclast cytoskeletal boundaries with Phallodin-Alexa Flour 488 (Thermo Fisher Scientific, 1:2000) and nuclei with Hoechst 33342 (Thermo Fisher Scientific, 1:5000). Cells were fixed as described above, washed with PBS, and stained with toxin/dyes for 1 hr in complete staining buffer (PBS + 5% FBS + 0.1% TX100) before a final PBS wash prior to imaging. Ten selected fields of view were imaged on a grid from the center of each well/dish in automated fashion using Alexa 488, Hoechst and phase contrast compatible filter cubes (BioTek) on a Lionheart FX microscope using a 10×/0.3 NA Plan Fluorite WD objective lens (BioTek) using Gen3.10 software (BioTek). Each image was separated by approximately 1800 μm in both x and y parameters.

All image data were evaluated using Fiji/ImageJ's open-source image processing package v.2.1.0/1.53c.

## Cell fusion quantitation

Osteoclast fusion efficiency was evaluated as the number of fusion events between cells in 10 images. In brief, since regardless of the sequence of fusion events, the number of cell-to-cell fusion events required to generate syncytium with $N$ nuclei is always equal to $N - 1$, we calculated the fusion number index as $\Sigma(N_i - 1) = N_{total} - N_{syn}$, where $N_i$ = the number of nuclei in individual syncytia and $N_{syn}$ = the total number of syncytia. We normalized the number of fusion events to the total number of nuclei (including unfused cells) to control for small variations in cell density from dish to dish and image to image. In contrast to traditional fusion index measurements, this approach gives equal consideration to fusion between two mononucleated cells, one mononucleated cell and one multinucleated cell and two multinucleated cells. In traditional fusion index calculations, fusion between two multinucleated cells does not change the percentage of nuclei in syncytia. If instead one counts the number of syncytia, a fusion event between two multinucleated is not just missed but decreases the number of syncytia. In contrast, the fusion number index is inclusive of all fusion events.

## Synchronization of osteoclast fusion

Osteoclast fusion was synchronized as described in *Verma et al., 2014*. Briefly, osteoclast media was refreshed with media supplemented with 100 ng/ml M-CSF, 100 ng/ml RANKL, and 350 μM lauroyl-LPC 72 hr post-RANKL treatment. Following 16 hr, LPC was removed via five washes with fresh media and cells were allowed to fuse in the presence or absence of antibody treatment or recombinant La at the concentrations described in the figure legends for 90 min.

## Transcript analysis

For real-time polymerase chain reaction (PCR), total RNA was collected from cell lysates using PureLink RNA kit following the manufacturer's instructions (Invitrogen # 12183018 A). cDNA was generated from total RNA via reverse transcription reactions using a High-Capacity RNA-to-cDNA kit according to the manufacturer's instructions (Applied Biosystems, # 4387406). cDNA was then amplified using the iQ SYBR Green Supermix (Bio-Rad). All primers were predesigned KiCqStart SYBR Green primers with the highest rank score specific for the gene of interest or glyceraldehyde 3-phosphate dehydrogenase (GAPHD) control and were used according to the manufacturer's instructions (Sigma). All real-time PCR reactions were performed and analyzed on a CFX96 real-time system (Bio-Rad), using GAPDH as an internal control. Fold-change of gene expression was determined using the ΔΔCt method. Three to four independent experiments were performed, and each was analyzed in duplicate.

| Gene | Forward primer sequence | Reverse primer sequence |
|---|---|---|
| NFATc1 | catttcggaatcagaggataac | ttataattggaacgttggcg |
| cFOS | cagttatctccagaagaagaag | cttctagttggtctgtctcc |
| SSB | gagcaaaagaggggataattc | Ccttctagtacttcccaagtc |
| ANXA5 | attaagggagatacatctggg | gcatgctagtatgaataaggc |
| GAPDH | acagttgccatgtagacc | ttgagcacagggtacttta |

## Mineral resorption

Mineral resorption was evaluated using mineral resorption assay kits from Cosmo Bio USA according to the manufacturer's instructions. In short, fluoresceinamine-labeled chondroitin sulfate was used to label 24-well, calcium phosphate-coated plates. Human, monocyte-derived osteoclasts were differentiated as described above, using α-MEM without phenol red. Media were collected at 4–5 days post-RANKL addition, and fluorescence intensity within the media was evaluated as recommended by the manufacturer. Data were normalized to the level of fluorescence released by control cells where RANKL was not added.

## Statistical analysis

Statistical analyses were performed using Prism software (GraphPad Prism version 8.0.0). Unless stated in the legend, differences between groups were observed in each experiment, cells from each donor were paired across the conditions described, and statistical significance was assessed via Student's $t$-test. Due to the inherent variability in the derivation of primary human monocytes to osteoclast, we analyzed statistical significance using a ratio paired $t$-test, where the raw values for the assay are logarithmically transformed and then assessed, when the precise time course of osteoclast differentiation and baseline extents of fusion varied considerably from donor to donor. The quantitated results presented all represent the mean ± the standard error of the mean. While the p values for each statistical comparison are defined in the legends of each figure, we graphically represented our statistical evaluations using the following symbols: ns, $p = >0.05$; *$p = <0.05$; **$p = <0.01$; ***$p = <0.001$.

## Acknowledgements

LVC thanks Dr. Alexander Peskin, University of Otago for enjoyable discussion. We thank the National Institutes of Health Department of Transfusion Medicine for isolating the monocytes used in this study. Our work was supported by the Intramural Research Program of the Eunice Kennedy Shriver National Institute of Child Health and Human Development, National Institutes of Health and by the Office of Research on Women's Health (ORWH) through the Bench to Bedside Program award # 884515.

## Additional information

### Funding

| Funder | Grant reference number | Author |
|---|---|---|
| Eunice Kennedy Shriver National Institute of Child Health and Human Development | 1k99-HD110609-01 | Jarred M Whitlock |
| Eunice Kennedy Shriver National Institute of Child Health and Human Development | Intramural Research Program | Leonid Chernomordik |
| Office of Research on Women's Health | 884515 | Leonid Chernomordik |

| Funder | Grant reference number | Author |
|---|---|---|

The funders had no role in study design, data collection, and interpretation, or the decision to submit the work for publication.

## Author contributions
Evgenia Leikina, Formal analysis, Investigation, Methodology, Writing – review and editing; Jarred M Whitlock, Conceptualization, Data curation, Formal analysis, Supervision, Investigation, Methodology, Writing – original draft, Project administration, Writing – review and editing; Kamran Melikov, Conceptualization, Data curation, Writing – review and editing; Wendy Zhang, Data curation, Formal analysis, Writing – review and editing; Michael P Bachmann, Methodology, Writing – review and editing; Leonid Chernomordik, Conceptualization, Data curation, Supervision, Methodology, Writing – original draft, Project administration, Writing – review and editing

## Author ORCIDs
Jarred M Whitlock ⓘ https://orcid.org/0000-0002-5886-2047
Wendy Zhang ⓘ https://orcid.org/0009-0006-9245-1184
Leonid Chernomordik ⓘ http://orcid.org/0000-0001-7131-9244

## Ethics
Elutriated monocytes from healthy donors were obtained through the Department of Transfusion Medicine at National Institutes of Health under protocol 99-CC-0168 approved by the National Institutes of Health Institutional Review Board. Research blood donors provided written informed consent and blood samples were de-identified prior to distribution, Clinical Trials Number: NCT00001846. We also used elutriated monocytes from healthy donors obtained through Elutriation Core Facility, University of Nebraska Medical Center, informed consent was obtained under an Institutional Review Board approved protocol for human subject research 0417-22-FB. Research blood donors provided informed consent and samples were de-identified prior to distribution.

Reviewer #1 (Public Review): https://doi.org/10.7554/eLife.98665.3.sa1
Reviewer #2 (Public Review): https://doi.org/10.7554/eLife.98665.3.sa2
Author response https://doi.org/10.7554/eLife.98665.3.sa3

# Additional files

## Supplementary files
• MDAR checklist

## Data availability
All summary, tabular data in both normalized and raw formats are included in the source data files linked to each figure.

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
