## [Editor Report · eLife assessment]

This manuscript provides an **important** advance in our understanding of the molecular events that promote osteoclast fusion. **Compelling** data support the conclusion that an oxidized form of the ubiquitous protein La promotes osteoclast fusion following enrichment at the cell surface of osteoclast progenitors. These data improve our understanding of the processes that regulate bone resorption and will be of broad interest to researchers in the fields of cell biology and musculoskeletal physiology.

---

## [Referee Report · Reviewer #1 (Public Review)]

In this manuscript, Leikina et al. investigate the role of redox changes in the ubiquitous protein La in promotion of osteoclast fusion. In a recently published manuscript, the investigators found that osteoclast multinucleation and resorptive activity are regulated by a de-phosphorylated and proteolytically cleaved form of the La protein that is present on the cell surface of differentiating osteoclasts. In the present work, the authors build upon these findings to determine the physiologic signals that regulate La trafficking to the cell membrane and ultimately, the ability of this protein to promote fusion. Building upon other published studies that show (1) that intracellular redox signaling can elicit changes in the confirmation and localization of La, and (2) that osteoclast formation is dependent on ROS signaling, the authors hypothesize that oxidation of La in response to intracellular ROS underlies the re-localization of La to the cell membrane and that this is necessary for its pro-fusion activity. The authors test this hypothesis in a rigorous manner using antioxidant treatments, recombinant La protein, and modification of cysteine residues predicted to be key sites of oxidation. Osteoclast fusion is then monitored in each condition using fluorescence microscopy. These data strongly support the conclusion that oxidized La is de-phosphorylated, increases in abundance at the cell surface of differentiating osteoclasts, and promotes cell-cell fusion. A strength of this manuscript is the use of multiple complementary approaches to test the hypothesis, especially the use of Cys mutant forms of La to directly tie the observed phenotypes to changes in residues that are key targets for oxidation. The manuscript is also well written and describes a clearly articulated hypothesis based on a precise summation of the existing literature. The findings of this manuscript will be of interest to researchers in the field of bone biology, but also more generally to cell biologists. The data in this manuscript may also lead to future studies that target La for bone diseases in which there is increased osteoclast activity. Weaknesses of the first version of the manuscript were minor and predominantly related to data presentation choices and some statistical analyses. These weaknesses were comprehensively addressed in the revised manuscript, and therefore the study has increased clarity and rigor.

---

## [Referee Report · Reviewer #2 (Public Review)]

Summary:

Bone resorption by osteoclasts plays an important role in bone modeling and homeostasis. The multinucleated mature osteoclasts have higher bone-resorbing capacity than their mononuclear precursors. The previous work by authors has identified that increased cell-surface level of La protein promotes fusion of mononuclear osteoclast precursor cells to form fully active multinucleated osteoclasts. In the present study, the authors further provided convincing data obtained from cellular and biochemical experiments to demonstrate that the nuclear localized La protein where it regulates RNA metabolism was oxidized by redox signaling during osteoclast differentiation and the modified La protein was translocated to osteoclast surface where it associated with other proteins and phospholipids to trigger cell-cell fusion process. The work provides novel mechanistic insights into osteoclast biology and provides a potential therapeutic target to suppress excessive bone resorption in metabolic bone diseases such as osteoporosis and arthritis.

Strengths:

Increased intracellular ROS induced by osteoclast differentiation cytokine RANKL has been widely studied in enhancing RANKL signaling during osteoclast differentiation. The work provides novel evidence that redox signaling can post-translationally modify proteins to alter the translocation and functions of critical regulators in the late stage of osteoclastogenesis. The results and conclusions are mostly supported by the convincing cellular and biochemical assays,

Weaknesses:

Lack of in vivo studies in animal models of bone diseases such as postmenopausal osteoporosis, inflammatory arthritis, and osteoarthritis reduces the translational potential of this work.

---

## [Author Response]

The following is the authors’ response to the original reviews.

**Recommendations for the authors:**

**Reviewer #1 (Recommendations For The Authors):**
(1) When introducing the different antibody clones recognizing Pan, oxidized, or reduced forms, please clearly indicate which clone number belongs to which form.

- We see where the original language could be confusing. Please see our new introduction to the antibodies used.

“we evaluated the redox state of La in fusing osteoclasts using recently validated monoclonal α-La antibodies that recognize oxidized La (clone 7B6) or reduced La (clone 312B), or do not distinguish between these La species (Pan, clone 5B9)”

(2) "Finding that the surface La pool, which promotes multinucleation in osteoclasts, is an oxidized species..." I would suggest rewording as "...is enriched in oxidized species".

- Agreed. We have edited the sentence as follows.

“Finding that the surface La pool, which promotes multinucleation in osteoclasts, is enriched in an oxidized species raised the question”

(3) Although not necessary to support the conclusions of the manuscript, it would be interesting to know if the application of La194-408 to osteoclast progenitors following NAC treatment results in the rescue of La staining at the cell surface, or if this exogenous La is acting independently from cell surface association.

- We agree that this is an interesting idea. We previously demonstrated that we could add La 1-375 to osteoclast progenitors following RANKL addition and promote osteoclast fusion. We also demonstrated that La 1-375 under these conditions enriched La surface staining (PMID: 36739273)

- Therefore, we hypothesize that La 194-408 would act similarly.

(4) Is the confirmation of La modified by the conversion of Cys 232 and 245 to alanine? What about the potential to form oligomers?

- To directly answer the Reviewer’s question – we simply do not know and do not have a simple way to test this. To speculate, the differential recognition of La that is reduced vs oxidized by the antibodies used here (specifically clone 312b vs clone 7b6) suggests that some conformational change is taking place when redox signaling modifies La in osteoclasts. Moreover, in Supp. Fig. 4b, we show that recombinant La 194-408 does form a small amount of dimer under our conditions while La 194-408 Cys 232 and 245 to Ala does not. These data together weakly support that La, when converted from reduced to oxidized forms or when we artificially Cys 232 and 245 to Ala, undergoes some conformational and oligomeric change. However, we are not comfortable making

such claims in the manuscript currently and prefer to investigate this with more rigor and comment in the biological significance of these potential changes in the future.

(5) "In conclusion, in this study, we identified redox signaling as a molecular switch that redirects La protein away from the nucleus, where it protects precursor tRNAs from exonuclease digestion, and towards its osteoclast-specific function at the cell surface..." I would suggest rewording this sentence given that there is no evidence that the function of oxidized La at the cell surface is osteoclast-specific. This phenomenon could be applicable to other cell types and other biological processes.

- The Reviewer makes a good point here, that we very much appreciate. We hoped to communicate that this was a unique function of La that was different from the well-recognized role this protein plays in RNA metabolism, but somewhat overstated past our intention. Please see where we have modified this statement to read:

“In conclusion, in this study, we identified redox signaling as a molecular switch that redirects La protein away from the nucleus, where it protects precursor tRNAs from exonuclease digestion, and towards its separable function at the osteoclast surface, where La regulates the multinucleation and resorptive functions of these managers of the skeleton.”

(6) In methods, the definition of TCEP is missing a closed parenthesis sign.

- Thank you, corrected.

(7) In methods under "Cells" there is a missing superscript in 1x106 cells/ml. Presumably, this is 1x10e6.

- Thank you, corrected.

(8) Please provide the sequences of primers used for RT-PCR in this study.

- Understood. Please see where a table of all primer sequences used has been added to the Methods under the Transcript Analysis section.

(9) In methods, "Bone resorption" should be relabeled given that the osteoclasts are plated on calciumphosphate plates and not on a bone surface.

- Thank you. Please see where in the Methods both the title and all references to “bone resorption” in the method description have now been changed to “mineral resorption”.

(10) In several figures, it would be more appropriate to correct for multiple comparisons in the statistical analyses.

- We appreciate this concern. Please see where Fig. 2b,c; Fig. 3 b,c; Fig. 4d; Fig. 5b,d; and Fig. 6d have been reanalyzed using paired one-way ANOVAs corrected for multiple comparisons. Now all data where t-tests are used to evaluate statistical significance are only evaluating differences between 2 values and all experiments considering 3+ values are compared using one-way ANOVAs corrected for multiple comparisons.

(11) Figure 5: Panels D and E are flipped relative to the legend. Please also define the reagent used for ROS signal in the legend.

- Thank you. D and E are now corrected and we added “(Grey = CellRox Dye)” to the end of the legend for Fig. 5a.

(12) Supplemental Figure 5c: in the control condition, why are some nuclei not staining with the reduced La antibody?

- Great question, direct answer – we simply do not know.

Longer answer, this image is in fact representative and not exclusive to the reduced La antibody (clone 312b). When we look at La staining in mature, multinucleated osteoclast nuclei at later timepoints post fusion using even pan antibodies, we find that its localization to the nuclei of syncytial osteoclasts is not uniform, but that nuclear La preferentially enriches in some mature osteoclast nuclei and seems to be excluded from others. This may suggest that – akin to myonuclei in skeletal muscle – osteoclast nuclei in a syncytium are not all equal. However, we are far, far away from being able to make any conclusions from the data we have.

(13) Figure 7 legend: consider breaking this legend up into multiple sentences.

- Thank you for the suggestion. The legend for Figure 7 has been rewritten.

**Reviewer #2 (Recommendations For The Authors):**
(1) Can the authors use the official name of La protein in NCBI GENE and PROTEIN?

- While some in the field refer to lupus La protein as La protein, we choose to refer to it simply as La, as is common throughout the Lupus La Protein literature. It is our opinion that continuously referring to a protein as a name + the word protein throughout the manuscript is unnecessary and alters the flow of our manuscript’s points.

Thanks. We have included the official name of human La in NCBI GENE (SSB small RNA binding exonuclease protection factor La, Gene ID 6741, NCBI GENE) into the revised text.

(2) The references 26 and 27 are not representative. The pioneering work from Mundy, Chambers, and Almeida (PBMID 2312718, 15528306, and 24781012) should be cited.

- Thanks. We have added these 3 references to better acknowledge these significant contributions.

(3) It is hard to understand Figure 2. What are the white arrows in Figure 2a pointed to? In Figure 2b, what do the columns a-LA(Red), a-La (Pan), and a-La (Ox) mean, treatment, or staining? Figure 2c, the legend "conditions where surface proteins are oxidized (TCEP) seems to be "deoxidized.

- We agree. We now realized this legend was rather confusing. It has been edited to read

“(a) Representative fluorescence and DIC confocal micrographs of primary human osteoclasts following synchronized cell-cell fusion where hemifusion inhibitor was left (Inhibition), removed (Wash) or removed but the α-La antibodies indicated were simultaneously added.

Cyan=Hoechst Arrows=Multinucleated Osteoclasts (b) Quantification of a.” • Thanks. 2c has now been corrected to “reduced” rather than the errant “oxidized”.

(4) How do authors normalize bone resorption, % of total area?

- We normalized to a separate, paired well where monocytes are differentiated to precursors (MCSF), but no RANKL is added. We have added this omitted information to the methods sections for our mineral resorption assay.

(5) Figure 5. There are two legends (b). In Figure 5c RT-qPCR, the DC-STAMP or OC-STAMP and mature osteoclast marker calcitonin receptor should be included.

- Thank you. There were several problems with Figure legend 5 that both you and Reviewer #1 brought our attention to. We have now corrected these errors.

- We understand the Reviewer’s interest in these markers. However, our point is that the steadystate transcript levels of two well recognized osteoclast differentiation factors and the fusion regulator La, which our manuscript focuses on, are not significantly altered by NAC treatment at these later, fusion associated timepoints. While DC-STAMP, OC-STAMP, and Calcitonin would be interesting, we believe they are outside the scope of this manuscript.